# Consequences of Metabolic Interactions during *Staphylococcus aureus* Infection

**DOI:** 10.3390/toxins12090581

**Published:** 2020-09-09

**Authors:** Tania Wong Fok Lung, Alice Prince

**Affiliations:** Department of Pediatrics, Vagelos College of Physicians & Surgeons, Columbia University, New York, NY 10032, USA; tw2595@cumc.columbia.edu

**Keywords:** *Staphylococcus aureus*, small colony variants (SCVs), biofilm, metabolic adaptation, metabolic reprogramming, cell death, epigenetics, trained immunity, chronic or persistent infection

## Abstract

*Staphylococcus aureus* is a metabolically flexible pathogen that causes infection in diverse settings. An array of virulence factors, including the secreted toxins, enables *S. aureus* to colonize different environmental niches and initiate infections by any of several discrete pathways. During these infections, both *S. aureus* and host cells compete with each other for nutrients and remodel their metabolism for survival. This metabolic interaction/crosstalk determines the outcome of the infection. The reprogramming of metabolic pathways in host immune cells not only generates adenosine triphosphate (ATP) to meet the cellular energy requirements during the infection process but also activates antimicrobial responses for eventual bacterial clearance, including cell death pathways. The selective pressure exerted by host immune cells leads to the emergence of bacterial mutants adapted for chronicity. These host-adapted mutants are often characterized by substantial changes in the expression of their own metabolic genes, or by mutations in genes involved in metabolism and biofilm formation. Host-adapted *S. aureus* can rewire or benefit from the metabolic activities of the immune cells via several mechanisms to cause persistent infection. In this review, we discuss how *S. aureus* activates host innate immune signaling, which results in an immune metabolic pressure that shapes *S. aureus* metabolic adaptation and determines the outcome of the infection.

## 1. Introduction

*Staphylococcus aureus* is a commensal organism, colonizing the skin and nares of 30% of the population, and can cause a range of infections with different clinical severities. Its ubiquity both as a component of the normal flora and as a pathogen reflects a substantial degree of metabolic flexibility, as well as the ability to elude immune clearance. Infection is associated with metabolic changes in both the host and microorganism: in the host to fuel an immune response [1] and in the bacteria to synthesize the gene products necessary to elude phagocytic clearance and to generate adenosine triphosphate (ATP) for protein synthesis and survival [2]. Over the course of infection, the metabolic activities of both host and pathogen are dynamically regulated, reflecting the requirements for an acute or a chronic infection.

Host immune cells rapidly adapt to a perceived infectious threat by altering metabolism from oxidative phosphorylation (OXPHOS), the typical pathway utilized in resting macrophages, to glycolysis [3,4,5,6]. Increased glycolysis and the accompanying breakpoint in the tricarboxylic acid (TCA) cycle at succinate dehydrogenase result in succinate accumulation, which in turn stabilizes the transcription factor hypoxia-inducible factor-1α (HIF-1α) and increases interleukin-1β (IL-1β) expression, linking the metabolic and immune responses to infection [7] (Figure 1A). Thus, the host diverts the utilization of ATP from cell maintenance activities to self-defense modalities with the generation of cytokines. Bacteria, similarly, must respond to a new environmental niche and the onslaught of immune cells and their products. *S. aureus* is especially responsive to the setting of acute infection and alters its own gene expression in response to host metabolic changes [8]. Such transcriptional changes serve to promote chronic infection.

Exactly what host factors drive the adaptation of *S. aureus* to the host have not been fully delineated, but they include immune and metabolic pressures. Conditions at specific sites of infection are important in the selection of *S. aureus* variants that are able to maintain infection [6,9]. Toxins that are critical in enabling invasion across epithelial or endothelial barriers may become dispensable and even detrimental, once a chronic infection is established. The bacterial metabolic requirements at a mucosal site may differ substantially from those once the bacteria have reached the bloodstream. Such adaptive changes may be genetic and correlate with the accumulation of single nucleotide polymorphisms (SNPs) in strains harvested over time from sites of chronic infection [3,10,11]. Over the course of infection, clinical isolates of *S. aureus* often have altered expression of toxins [12] that is associated with mutations in central regulatory genes such as the *agr*, *sar* and *codY* loci [11,13,14]. Perhaps more commonly, adaptive change is at the level of transcriptional regulation, enabling the microorganisms to rapidly adapt to different metabolic modes, as reflected by planktonic versus sessile/biofilm lifestyles.

The simultaneous changes in host and staphylococcal metabolism during *S. aureus* infection have a major impact on the clearance of infection [6,9,15,16]. In the pathogenesis of pneumonia, aspirated bacteria must rapidly adjust their metabolic activity to consume nutrients now available in the airway that differ from those in the sites of initial colonization in the nares. However, airway epithelial cells and recruited immune cells perceive competition for nutrients driving changes in host metabolic activity. In this review, we will focus on the host response to *S. aureus*, highlighting the importance of the activation of immunometabolic responses and the role of toxins in promoting the establishment of infection through the induction of host cell death pathways such as pyroptosis and necroptosis. We will also review how these alterations in host metabolism promote the selection of *S. aureus* variants best suited for chronic infection.

## 2. Host Responses to Acute Infection

Skin and soft tissues are the most common sites of *S. aureus* infection. The human skin, which constitutes the first line of defense against infection, is composed of multiple layers including the epidermis and the underlying dermis. The epidermis is itself organized into several layers, with the major component being keratinocytes. *S. aureus* is a common colonizer of damaged skin, as typical of atopic dermatitis [17]. In order to establish infection, *S. aureus* uses several virulence factors, especially toxins, phenol-soluble modulins (PSMs), adhesins and proteases [18]. These activate several cell death modalities, as discussed below, that contribute to barrier penetration and dissemination into underlying layers of the skin and promote bacterial infection.

### 2.1. Inflammasome Activation and Pyroptosis

The activation of a robust inflammatory response is one of the hallmarks of *S. aureus* infection in the skin. This includes the activation of the NLRP3 inflammasome, which is composed of two signals [19,20]. The first signal consists of NF-κB-mediated upregulation of pro-IL-1β transcription via sensing of lipoteichoic acid (LTA) by toll-like receptor 2 (TLR2), and the second signal involves caspase-1 activation, cleavage of pro-IL-1β and gasdermin D (GSDMD). In contrast to macrophages that require sensing of internalized peptidoglycan to initiate pro-IL-1β production [21], keratinocytes constitutively express pro-IL-1β and are primed for inflammasome activation [22]. Active caspase-1 cleaves GSDMD, leading to its oligomerization in the host cell membrane and the induction of pyroptotic cell death [20] (Figure 1A). Cleavage of pro-IL-β to its active form (signal 2) by active caspase-1 and its release from the cell are critical for the recruitment of polymorphonuclear leukocytes (PMNs) and eventual staphylococcal clearance, thereby contributing to tissue damage by pyroptosis (Figure 1A). Caspase-1 also activates calpains, Ca^2+^-dependent intracellular proteases, by targeting and degrading the calpain inhibitor calpastatin, promoting pyroptosis [22].

*S. aureus* activates the NLRP3 inflammasome activation via the pore-forming toxins (PFTs) including α-hemolysin (Hla), HlgAB, HlgCB, LukAB, LukED and LukSF/Panton–Valentine leucocidin (PVL) [23,24,25,26,27]. The expression of toxins in *S. aureus* is largely controlled by the accessory gene regulator (Agr) quorum sensing system, which mediates a switch from the expression of host matrix binding proteins to that of toxins at high bacterial cell densities [28]. These toxins, particularly Hla, activate a brisk host response, consisting of IL-1β production [22]. It has long been appreciated that IL-1β is a critical host factor in the pathogenesis of cutaneous *S. aureus* infection [29] and has a major role in *S. aureus* killing and clearance [29,30]. Mice deficient in IL-1β (*Il1-**β^−/−^*) developed larger lesions with higher bacterial burdens and reduced neutrophil recruitment following subcutaneous injection compared to parental strain mice [29]. Recombinant active IL-1β and adoptive transfer of IL-1β-expressing neutrophils helped control infection and enhanced bacterial clearance [29]. However, activation of the NLRP3 inflammasome is also associated with epithelial damage. Hla-deficient *S. aureus* but not PVL or protein A (SpA)-deficient strains reduced caspase-1 activation in keratinocytes [22]. The Hla-deficient mutants were less capable of transmigrating across human keratinocytes grown on transwells, unlike the WT strain or SpA- or PVL-mutants, consistent with significantly decreased dextran permeability from apical to basal chamber of transwells [22]. Thus, Hla-induced pyroptosis contributes substantially to the disruption of keratinocyte barrier integrity, enabling staphylococcal dissemination into deeper layers of skin [22]. Keratinocyte death by pyroptosis exposes underlying matrix materials, readily identified by the *S. aureus* microbial surface components recognizing adhesive matrix molecules (MSCRAMMs), and a nidus of infection is established.

The invasive ability of *S. aureus* is, in part, dependent upon the changes in the host. Transmigration of WT *S. aureus* was inhibited when keratinocytes were treated with either a caspase-1 inhibitor or a calpain inhibitor, indicating that while induction of the inflammasome and pyroptosis are important for staphylococcal clearance, they initially contribute to the initial establishment of staphylococcal infection. Of note, *S. aureus* Hla also stimulates epithelial proliferation via epidermal growth factor receptor (EGFR), activating host repair processes and proliferation [31]. Therefore, keratinocyte infection does not result in widespread loss of barrier function but instead results in focal areas of infection and induction of more keratinocytes to replace the necrotic cells as demonstrated in human organotypic cultures [22].

### 2.2. Consequences of S. aureus-Induced Necroptosis

In addition to activating pyroptosis, *S. aureus* also activates a caspase-independent form of cell death termed necroptosis [32,33,34] that instead relies on the formation of a complex composed of several host proteins including receptor interacting protein kinase 1 (RIPK1), toll/IL-1R domain containing adapter-inducing IFN-β (TRIF) or Z-DNA-binding protein 1 (ZBP1/DAI) that contain the RIPK homology interaction motif (RHIM). This complex mediates the recruitment and activation of RIPK3 and the necroptosis executioner mixed lineage kinase domain-like (MLKL) protein upon ligand binding. Active MLKL subunits oligomerize at the host cell membrane, inducing membrane rupture. *S. aureus* toxins can induce necroptosis in a variety of cell types including macrophages [32,33,34]. Although *S. aureus* activates necroptosis, this caspase-independent cell death does not eradicate *S. aureus* [6,34]. This is in contrast to pyroptosis. Of note, *S. aureus* activation of RIPK3 can also stimulate inflammasome activation and IL-1β production and secretion, which promote *S. aureus* clearance [6,34].

Exactly how *S. aureus* stimulates host cell death is important to the bacteria, as they are killed by inflammasome-associated pyroptosis but not by the RIPK-mediated necroptotic pathway [6,22,34]. This may be one of the factors that promote the emergence of the small colony variants (SCVs). SCVs, which are often associated with chronically infected tissues [35,36], were first characterized by Proctor et al. [37] as phenotypically distinct from other *S. aureus* colonies. SCVs are slow growing, have pinpoint colony size and a variety of mutations typically in genes associated with electron transport chain components such as menadione and heme [38]. As a result of these mutations, SCVs often have altered metabolism with decreased TCA cycle activity and OXPHOS and increased glycolysis to meet their energy requirements [39]. Despite their usual downregulation of toxin production, SCVs are associated with significant morbidity [36].

In contrast to the toxin-associated pathology associated with pyroptosis, the relatively indolent SCVs induce necroptosis [6] similarly to toxin-producing *S. aureus*. In cutaneous infection *S. aureus*-induced necroptosis is accomplished through metabolic crosstalk between the staphylococci and keratinocytes, regardless of the presence of toxins [6]. Staphylococcal glycolysis, which is required for skin infection, stimulates metabolic competition for glucose with the host cells [5] (Figure 1A,B). This competition induces host glycolysis that generates formation of reactive oxygen species (ROS), which induce necroptosis [6] (Figure 1B). A prototypic SCV (Δ*hemB*) and the wild type strain stimulated keratinocyte glycolysis and necroptosis, unlike their heat-killed counterparts [6]. Necroptosis could be blocked by treatment of the infected keratinocytes with an inhibitor of glycolysis or an ROS scavenger [6]. Of note, stimulation of primary human keratinocytes with *S. aureus* mutants lacking toxin production (Δ*hla*, Δ*agr*, Δl*ukAB/ED/SF/hlgABC*) also induced necroptosis, indicating that the ability of *S. aureus* to perform glycolysis and in turn induce host glycolysis, was the defining factor for the stimulation of keratinocyte necroptotic cell death.

The induction of necroptosis by the SCVs enabled these strains to cause persistent infection. Although cutaneous infection of mice with the prototypic SCV was more indolent than infection with the WT strain, the SCVs were able to persist, and both *Mlkl**^−/−^* and *Ripk1*^D138N/D183N^ (kinase inactive) mice that are unable to undergo necroptotic cell death had significantly fewer SCVs at the site of infection compared to wild type mice [6]. Metabolic stimuli, even when staphylococcal toxin production is reduced, activate host cell death pathways which do not necessarily eliminate staphylococci, and promote the selection of toxin-negative mutants that fail to activate the inflammasome.

## 3. Adaptive Metabolic Changes during Chronic *S. aureus* Infection

The activation of an acute host immune response, especially one dominated by IL-1β, initiates the recruitment and activation of phagocytes to clear *S. aureus*. This is especially important in the initial stages of acute infection. These immune cells ingest the bacteria and release ROS and elastase in response. *S. aureus* expresses numerous antioxidants, enabling some bacteria to withstand phagocytic killing [40]. It has long been thought that selection of *S. aureus* strains in vivo, is driven primarily through the selective pressure of immune cells [41]. This seems likely, given the large number of *S. aureus* gene products that are devoted to evading host immune clearance. Antiphagocytic protein A binds the Fcγ domain of immunoglobulin (Ig) and crosslinks the Fab domain of B cell receptors, blocking antibody development [42]. Antioxidant enzymes such as superoxide dismutase, catalase and staphyloxanthin defend *S. aureus* from host killing by reactive oxidants [40], and toxins selectively lyse human immune cells and many other immune-specific gene products [43].

Many *S. aureus* infections are more indolent and involve a subpopulation of microorganisms selected by the conditions within the host [6,9,16]. The in vivo selection of *agr*-deficient strains in clinical settings [44,45] is associated with the selection of strains lacking toxin expression. This is thought to prevent further recruitment of phagocytes [46]. In pulmonary and blood isolates of *S. aureus, agr* mutants have been identified and associated with poor clinical outcomes despite their lack of toxin expression [36,47]. A study of *S. aureus* strains from children with atopic dermatitis and chronic staphylococcal skin infection did not reveal many strains with *agr* mutation, and the expression of toxins was highly variable, as was their induction of cytokines [3]. Surveys of clinical strains from a variety of sites indicate that there are relatively few loss-of-function SNPs in major virulence genes, as might be expected, but many strains had accrued mutations or changes in the levels of expression of metabolic genes [3,10,11,12].

Significantly increased expression of *fumC* has been noted in several collections of clinical isolates [3,10]. FumC, a fumarate hydratase which converts fumarate to malate, was upregulated by almost 100,000-fold in isolates from skin infections [3] and is also significantly elevated in pulmonary isolates of *S. aureus* from patients with cystic fibrosis [10]. The increased *fumC* response is present in the SCV prototype, Δ*hemB* mutant, which is auxotrophic for hemin [6], suggesting that the control of fumarate accumulation is important for staphylococcal persistence. The increased expression of *fumC* has several benefits for the bacteria, including promoting glycolysis, which is indispensable during skin infection [5]. In addition, FumC, through the degradation of local fumarate, inhibits host trained immunity [6] (Figure 2A). Trained immunity refers to increased protection against a secondary infection at the site of the primary challenge. Fumarate, which is itself a glycolytic inhibitor [48], induces epigenetic changes in macrophages that promote trained immunity, enhancing cytokine production [49]. Fumarate inhibition of KDM5 histone demethylases promotes histone modifications such as histone H3 lysine 4 trimethylation (H3K4me3) at promoters of proinflammatory cytokines, serving to enhance transcription upon secondary insult [49]. Increased *fumC* expression by SCVs results in lower levels of fumarate during infection of human macrophage-like cells (THP-1 cells) and peripheral blood mononuclear cells (PBMCs) [6]. In vivo, fumarate degradation by the SCVs resulted in diminished protection from a secondary staphylococcal challenge in a mouse model of skin infection and promoted recurrent infection [6].

## 4. Host Metabolism Promotes Infection

It is increasingly apparent that the coordinated metabolic responses of both host and pathogen are involved in the initial and chronic responses to infection. As host glycolysis entails the release of succinate, stabilization of HIF-1α and IL-1β release, all expected to clear S. *aureus*, the metabolic activities that are beneficial for the bacteria, must outweigh this component of the immune response. In the skin, staphylococci maintain glycolysis, which is vital for their survival, by fumarate degradation. This metabolic balance is also observed in bone, one of the most frequent sites of invasive and chronic disease, in which glycolysis is also essential for *S. aureus* survival [9]. Of note, glutamate accumulates in infected bone, which competes with and inhibits exogenous aspartate acquisition by *S. aureus*. Thus, in this setting, S. *aureus,* despite expressing an aspartate transporter GltT, becomes dependent upon its own ability to synthesize aspartate [9] (Figure 2B). Other metabolic pathways such as the pentose phosphate pathway, TCA cycle and gluconeogenesis were, however, dispensable during staphylococcal bone infection. These findings indicate that the metabolic requirements of *S. aureus* are influenced by the local metabolic milieu and drive the selection of strains best suited for infection at specific sites.

The metabolic preferences of the monocytes recruited during the immune response to *S. aureus* also influence the outcome of the infection [16]. In contrast to the *S. aureus* isolates from the skin that induce host glycolysis and inflammation, the strains from prosthetic joint infections, which are prolific biofilm formers, promote oxidative phosphorylation in the recruited macrophage–monocyte population [16]. These immune cells have anti-inflammatory properties including IL-10 and arginase production. When their oxidative metabolism is blocked using a nanoparticle approach for the delivery of the OXPHOS inhibitor oligomycin, significantly increased inflammation and *S. aureus* clearance are observed [16]. The ability of biofilm formers to induce host IL-10 production was shown to be due to *S. aureus* lactate-mediated inhibition of the histone deacetylase 11 (HDAC11), resulting in unchecked HDAC6 activity, an increase in histone 3 (H3) acetylation at the *Il-10* promoter and enhanced *Il-10* transcription [50] (Figure 2C). These findings share some major host epigenetic features with the *fumC*-induced depletion of fumarate and suppression of trained immunity, discussed above, linking *S. aureus*-host metabolic adaptation and epigenetics during persistent infection.

## 5. Conclusions

Adaptive changes in both host and bacterial metabolism have profound effects on the pathogenesis of infection. The changing metabolic properties of the bacteria are intimately linked to the nature of the host response elicited. It has become evident that there are numerous mechanisms for the host to adjust its metabolic activity in response to *S. aureus* infection; these adjustments include the preferential utilization of specific metabolic cascades and may occur at the level of the mitochondria. The host, especially in response to toxin-producing *S. aureus*, can auto-destruct through the highly inflammatory inflammasome activation and pyroptosis. This immune pressure promotes *S. aureus* adaptation and the emergence of variants that are better suited to persist in a setting of accumulating oxidants by altering their transcriptional and metabolic profiles to evade pyroptotic cell death and clearance, as exemplified by the SCVs during cutaneous infection. The upregulation of *fumC* expression in SCVs and clinical *S. aureus* isolates is a newly discovered adaptive mechanism that confers several advantages to these host-adapted strains. Degradation of the glycolytic inhibitor fumarate by increased FumC ensures that these microorganisms can perform glycolysis, which is essential for *S. aureus* survival in various settings including the skin and bone. Fumarate is beneficial for host cells by inducing innate immune memory/trained immunity via the inhibition of histone demethylases. Inhibition of trained immunity via *fumC*-mediated depletion of fumarate promotes *S. aureus* SCV recurrence. Thus, the metabolic flexibility of *S. aureus* reflects its ability to adapt to or tolerate various microenvironments and cause chronic infections.

While it has long been thought that *S. aureus* persistence in clinical settings is associated with strains lacking toxin expression, numerous recent studies using clinical materials indicate that many isolates from chronic infections still express toxins. Many of these strains rewire their metabolism to promote biofilm production. *S. aureus* biofilm producers significantly influence the metabolic activity of the host by stimulating OXPHOS in macrophages and their production of the anti-inflammatory cytokine IL-10. Lactate produced by these biofilm-producing *S. aureus* inhibits host histone deacetylase HDAC11, resulting in enhanced IL-10 expression. Rewiring monocyte metabolism to a proinflammatory state via the delivery of an OXPHOS inhibitor in nanoparticles in vivo significantly reduced biofilm burdens. These examples highlight the role of metabolites in epigenetics and host immunity during staphylococcal infection and open a new avenue of investigation into therapeutic strategies that reprogram host immune metabolism to boost an effective host immune response and combat chronic infection. These strategies could also be combined with antimicrobial agents to enhance bacterial susceptibility to antibiotics and further improve immune clearance.

New therapeutic strategies are particularly important given that the majority of current strategies including vaccines focus on targeting staphylococcal toxins or surface adhesins. While these are undoubtedly highly effective against virulent *S. aureus* strains, they fail to target indolent strains such as the SCVs that arise as a result of metabolic adaptation to immune pressure and persist in the host. Thus, deepening our understanding of the immunological and metabolic consequences of *S. aureus* infection will help to design better therapeutic strategies to prevent chronic infections.

## Figures and Tables

**Figure 1 toxins-12-00581-f001:**
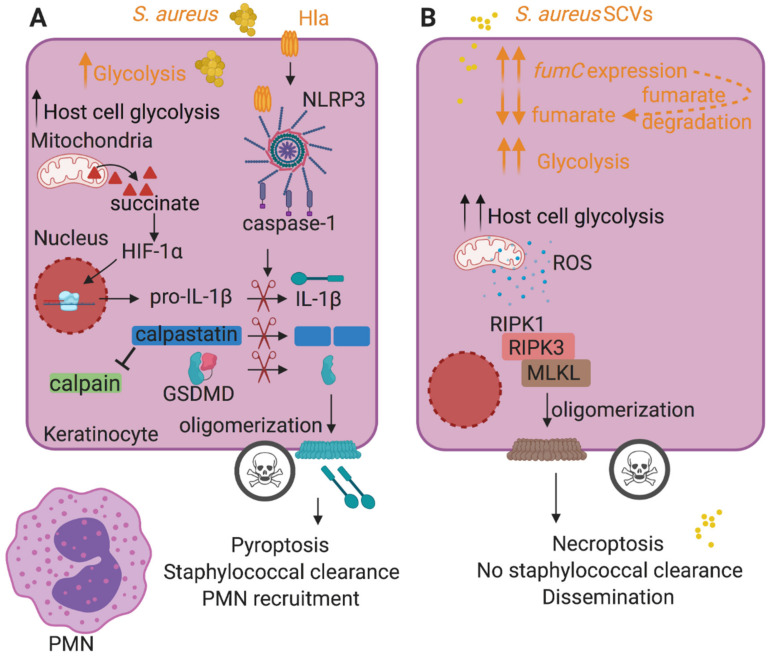
Host cell death pathways induced by *S. aureus* influence the outcome of infection. (**A**) During staphylococcal infection, bacteria and host cells compete for glucose, stimulating bacterial and host cell (e.g., keratinocyte) glycolysis. Host metabolic reprogramming results in succinate production and secretion. Succinate stabilizes the host transcription factor hypoxia-inducible factor-1α (HIF-1α) and increases pro-interleukin-1β (pro-IL-1β) expression. *S. aureus* toxins such as Hla activate the NLRP3 inflammasome, resulting in caspase-1 activation. Active caspase-1 cleaves pro-IL-1β into the mature form and gasdermin D (GSDMD) into N-terminal and C-terminal fragments. The N-terminal fragment of GSDMD oligomerizes at the cell membrane, forming lytic pores and inducing pyroptotic cell death that eventually recruits polymorphonuclear leukocytes (PMNs) and leads to bacterial clearance. Caspase-1 also cleaves calpastatin, relieving calpain inhibition and promoting pyroptosis. (**B**) *S. aureus* induces necroptosis independently of toxin production during keratinocyte infection. *S. aureus* small colony variants (SCVs), with downregulated toxins, stimulate necroptosis, a caspase-independent cell death modality that does not kill *S. aureus* and promotes bacterial dissemination, similarly to the WT strain. Their glycolytic nature stimulates keratinocyte glycolysis and promotes the production of reactive oxygen species (ROS) and necroptosis. SCVs sustain glycolysis by increasing *fumC* expression that promotes the degradation of fumarate, a glycolytic inhibitor. Host-related activities are shown in black font and staphylococcal activities are shown in orange font.

**Figure 2 toxins-12-00581-f002:**
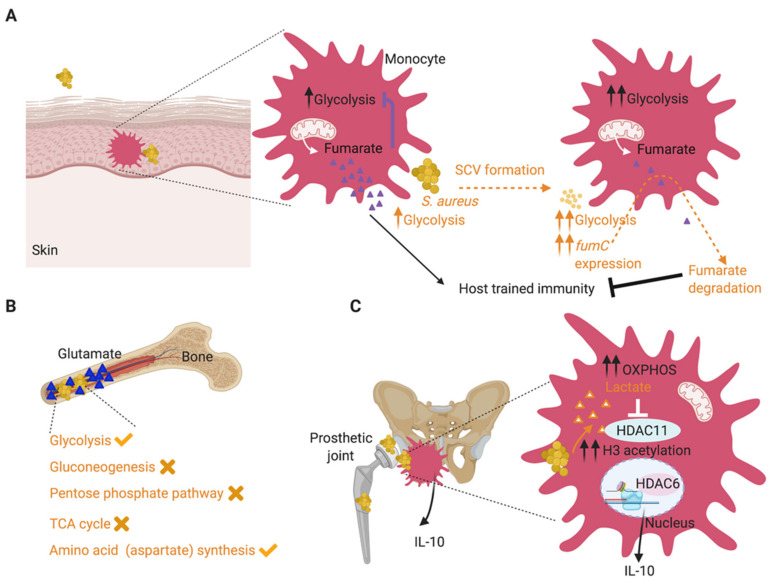
Adaptive metabolic changes during *S. aureus* infection promote chronic infection. (**A**). During skin infection, *S. aureus* SCVs adapt to local fumarate accumulation by overexpressing *fumC* to degrade it. This helps sustain glycolysis, given the role of fumarate as a glycolytic inhibitor. Fumarate degradation is also detrimental to the host and abrogates trained immunity, promoting recurrent infections. (**B**)**.** During staphylococcal bone infection, which is often chronic, *S. aureus* adapts to the excess glutamate in the infected tissues by stimulating glycolysis and aspartate biosynthesis, both critical pathways for staphylococcal survival in this milieu. Staphylococcal gluconeogenesis, pentose phosphate pathway and TCA cycle were dispensable for survival during osteomyelitis. (**C**)**.**
*S. aureus* biofilms stimulate a metabolic bias in recruited monocytes, favoring oxidative phosphorylation (OXPHOS) over glycolysis and facilitating their anti-inflammatory activity and biofilm persistence. This metabolic bias is stimulated by *S. aureus* biofilm-derived lactate, which promotes the production of anti-inflammatory IL-10, by inhibiting histone deacetylase HDAC11 and causing unchecked HDAC6 activity and increased histone acetylation at the *Il-10* promoter. Host-related activities are shown in black font and staphylococcal activities are shown in orange font.

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
