# Peer review of "Consequences of Metabolic Interactions during Staphylococcus aureus Infection"

_toxins, 2020, doi:10.3390/toxins12090581_

Round 1
Reviewer 1 Report
Manuscript “Activation of inflammatory and cell death pathways by S. aureus: consequences of metabolic adaptation” is a type of review publication that, based on the latest literature, analyzes the changes in metabolic activity, both in the cells of host and bacteria, during the acute and chronic infection process with S. aureus strains.
The topic is very interesting and important. Better knowledge of the immunological and metabolic consequences of S. aureus infection can help in the application of appropriate therapy and strategy to increase an effective immune response in chronic infections.The manuscript is divided into paragraphs logically arranged.The publication is enriched with two figures that facilitate understanding of the issues discussed. Conclusion: Reviewer recommends the presented manuscript for publication.
Author Response
We thank you for reviewing our manuscript and for your recommendation for publication.
Reviewer 2 Report
The topic of the review “activation of inflammatory and cell death pathways by Staphylococcus aureus: consequences of metabolic adaptation” is timely in light of the recent development in the field of immuno-metabolism in host-pathogen interactions.
This referee enjoyed reading this manuscript. However, they feel that it deserves to be amended and modified in some parts to improve readability and flow as the text is sometimes very patchy and not directing the reader to the point. Some sections are not guiding the authors towards the consequence of metabolic adaptation (in particular, apoptosis and autophagy are very poorly detailed and not appropriately illustrated).
General comments
The abstract of the proposed review should be modified to match what is expected from this vibrant field. The last sentence is not fitting what is expected from a review. Providing examples is per se the role of a review. Moreover, this is not a review exclusively on fumC, so no reason to single this pathway out. This referee strongly recommends the authors to reshape this abstract and articulate it to attract potential citations related to immuno-metabolism in host-pathogen interactions. Keywords unfortunately absent here.
1. The introduction:
Line 31: the abbreviation OxPhos should be introduced here.
The reference to OxPhos in resting macrophages should be supplemented with references relating to OxPhos in keratinocytes, osteoclasts and other S.aureus sites of infections (or at least discussed later in the manuscript).
Line 52-53: SCV don’t lack toxins. The genes are still there. The toxins genes are downregulated (as stated in line 66). SCVs deserve a better description than the one proposed here.
Paragraph 1.2 should be part of the introduction not standing alone.
Models of infection should be introduced in this section to help the reader understand that the sum of these observations is coming from different infection settings/models.
2. Inflammasome activation and pyroptosis:
This paragraph is a hodgepodge of toxins, and bacterial factors. It deserves a better structure and clearer message. Too many “to and fro” for such a small paragraph.
The mutant spa is mentioned here a fair bit. The authors are invited to describe the protein A activities on the host signalling pathways.
Line 102: The word traditionally is not adequate here.
3. The paragraphs on necroptosis (1.4) and SCV (1.5) should be merged together. Again, SCVs are not toxin deficient. They are non-toxinogenic. Until they revert to normal colony variants.
4. Points 1.6 and 1.7 should be a single section, and the title of that paragraph revised accordingly.
5. Point 1.8 should include comments and adequate references on S.aureus metabolic state during biofilm formation on inert and biological surfaces.
6. Point 1.9: Apoptosis and autophagy should be 2 different paragraphs. Considering how the other PCD have been detailed, apoptosis should be equally presented. Autophagy has not been discussed in the context of immuno-metabolism. This section is the poorest of the manuscript and should be worked on.
7. Unfortunately, the quality of figure 3 C is very poor. There are no indicators of what is described in the text. Where are the bacteria? Why did the author pick an agr mutant? What is observed with infection by the WT? Double autophagic membranes cannot be seen clearly. There are no scale bars, no lower magnification. In all, this figure does not bring much to the panel and could either be improved (if possible) or deleted.
8. Finally, the conclusion is too thin and offers poor perspectives on future areas of investigation in this important field.
Minor comments
- The bacterium’s name should be spelled out in the title and not abbreviated.
- The numbering of the section is not logical.
- Keywords are missing.
- The colours in the figures are too dim, the cytosol should have a colour different from aureus. Biorendeer legend should appear only once in the acknowledgements.
Author Response
The topic of the review “activation of inflammatory and cell death pathways by Staphylococcus aureus: consequences of metabolic adaptation” is timely in light of the recent development in the field of immuno-metabolism in host-pathogen interactions.
This referee enjoyed reading this manuscript. However, they feel that it deserves to be amended and modified in some parts to improve readability and flow as the text is sometimes very patchy and not directing the reader to the point. Some sections are not guiding the authors towards the consequence of metabolic adaptation (in particular, apoptosis and autophagy are very poorly detailed and not appropriately illustrated).
General comments
The abstract of the proposed review should be modified to match what is expected from this vibrant field. The last sentence is not fitting what is expected from a review. Providing examples is per se the role of a review. Moreover, this is not a review exclusively on fumC, so no reason to single this pathway out. This referee strongly recommends the authors to reshape this abstract and articulate it to attract potential citations related to immuno-metabolism in host-pathogen interactions. Keywords unfortunately absent here.
We thank the reviewer for his/her comments. We were remiss in failing to better present the recent developments in the field of immunometabolism and the resulting staphylococcal metabolic adaptation in the abstract. We have rewritten the abstract to better reflect the importance of metabolic interactions during infection. We apologize for omitting the keywords in the manuscript and have included them (lines 21-23).
- The introduction:
Line 31: the abbreviation OxPhos should be introduced here. We have added the abbreviation (now line 37).
The reference to OxPhos in resting macrophages should be supplemented with references relating to OxPhos in keratinocytes, osteoclasts and other S.aureus sites of infections (or at least discussed later in the manuscript). We have added references as requested (line 38, highlighted in yellow).
Line 52-53: SCV don’t lack toxins. The genes are still there. The toxins genes are downregulated (as stated in line 66). SCVs deserve a better description than the one proposed here. We completely agree with the reviewer that SCVs downregulate toxins and do not lack the toxin genes. We apologize for the confusion and have made the necessary changes (lines 81, 237). We would like to clarify that our description of the SCVs can be found in the text (lines 231-237), when they are first introduced in the context of their ability to induce necroptosis despite their downregulation of toxins, as opposed to in lines 80-81 (previously 52-53) which form part of the figure legend.
Paragraph 1.2 should be part of the introduction not standing alone. We apologize for the confusion. Paragraph 1.2 was not meant to stand alone and was supposed to lead into/include subsections 1.2.1 and 1.2.2, which were previously erroneously labelled as 1.3 and 1.4. We have made the necessary changes (lines 124, 126 and 215).
Models of infection should be introduced in this section to help the reader understand that the sum of these observations is coming from different infection settings/models. We have chosen to focus on the effect of cell death pathways in the skin given the importance of pyroptosis in S. aureus clearance in the skin and in promoting the emergence of bacterial strains that avoid the stimulation of this cell death modality. NLRP3 inflammasome activation and pyroptosis do not clear staphylococci from the lungs of infected mice given that WT and Nlrp3−/− mice had similar bacterial burdens following S. aureus intratracheal infection (Kebaier et al., J Infect Dis. (2012) 205:807–17). This has been discussed in our recent review “Pulmonary pathogens adapt to immune signaling metabolites in the airway” (Front Immunol. 2020; 11: 385).
- Inflammasome activation and pyroptosis:
This paragraph is a hodgepodge of toxins, and bacterial factors. It deserves a better structure and clearer message. Too many “to and fro” for such a small paragraph.
The mutant spa is mentioned here a fair bit. The authors are invited to describe the protein A activities on the host signalling pathways.
We apologize for the confusion. We have restructured this paragraph (lines 126-214) to better convey the importance of inflammasome activation and pyroptosis in not only bacterial clearance but also in the initial establishment of infection by exposing underlying layers of the epidermis upon pyroptotic cell death.
Line 102: The word traditionally is not adequate here. We have removed the word “traditionally”.
- The paragraphs on necroptosis (1.4) and SCV (1.5) should be merged together. Again, SCVs are not toxin deficient. They are non-toxinogenic. Until they revert to normal colony variants. These paragraphs were merged as requested (now paragraph 1.2.2, lines 215-343) and we have used the words “downregulation of toxins” for the SCVs (line 237).
- Points 1.6 and 1.7 should be a single section, and the title of that paragraph revised accordingly. These changes have been made as requested (now paragraph 1.3, lines 344-391).
- Point 1.8 should include comments and adequate references on S.aureusmetabolic state during biofilm formation on inert and biological surfaces. Whilst the metabolic state of S. aureus during biofilm on inert and biological surfaces is indeed very interesting, the main message/focus of point 1.8 (now point 1.4) is how the host metabolism promotes infection. To demonstrate this point, we have used the most recent example in the literature whereby S. aureus biofilm producers skew the host metabolic activity to OXPHOS to promote an anti-inflammatory environment that is conducive of chronic infection (Yamada et al. PLoS Pathog. 2020;16(3):e1008354).
- Point 1.9: Apoptosis and autophagy should be 2 different paragraphs. Considering how the other PCD have been detailed, apoptosis should be equally presented. Autophagy has not been discussed in the context of immuno-metabolism. This section is the poorest of the manuscript and should be worked on.
- Unfortunately, the quality of figure 3 C is very poor. There are no indicators of what is described in the text. Where are the bacteria? Why did the author pick an agr mutant? What is observed with infection by the WT? Double autophagic membranes cannot be seen clearly. There are no scale bars, no lower magnification. In all, this figure does not bring much to the panel and could either be improved (if possible) or deleted.
(for points 6 and 7) - We have removed apoptosis and autophagy from the review so as not to detract from the main recent developments in the field.
- Finally, the conclusion is too thin and offers poor perspectives on future areas of investigation in this important field.
We have expanded the conclusion to offer more perspectives on future areas of investigation.
Minor comments
- The bacterium’s name should be spelled out in the title and not abbreviated. We have made the necessary changes in the title.
- The numbering of the section is not logical. We have amended the numbering of the section in a more logical manner.
- Keywords are missing. We have included keywords.
- The colours in the figures are too dim, the cytosol should have a colour different from aureus. Biorendeer legend should appear only once in the acknowledgements. We have changed the color schemes and included BioRender only in the Acknowledgements section.
Reviewer 3 Report
This manuscript described the response of host inflammation and cell death path
This manuscript described the response of host inflammation and cell death pathway to S. aureus infection, focusing on the immunometabolic responses and the role of toxins in promoting the establishment of infection through the induction of host cell death pathways such as pyroptosis, necroptosis and apoptosis. The manuscript has summarized recent progress of host/pathogen’s metabolic adaption to different environments in different sections, complemented with illustrations by figures. For readers interested in S. aureus infections and adaption of SCVs, this is a very nice concise introduction of the complicated topics of host/pathogen immunometabloic response. There are few minor points of suggestion for improvement.
In figure 1B, it seems to indicate that SCVs stimulate necroptosis, in section 1.4, necroptosis can be also induced by S. aureus in general. Some clarification of figure 1 is needed.
Due to complexity of the topic with many cell types and pathways described in the text, it will be easier for readers to follow if a summary table is included with host cell types, pathways, major genes involved and references.
Abbreviation should be explained when first appeared in the text.way to S. aureus infection, focusing on the immunometabolic responses
Author Response
We thank the reviewer for his/her suggestions for improvement.
In figure 1B, it seems to indicate that SCVs stimulate necroptosis, in section 1.4, necroptosis can be also induced by S. aureus in general. Some clarification of figure 1 is needed. We have clarified both in the text (lines 238-330) and in the figure legend (lines 79-84) that both WT S. aureus and the SCVs which have downregulated toxin production are able to induce necroptosis in the skin and that this was instead dependent on the metabolic crosstalk between the bacteria and keratinocytes.
Due to complexity of the topic with many cell types and pathways described in the text, it will be easier for readers to follow if a summary table is included with host cell types, pathways, major genes involved and references. As pointed out by reviewer 2, we have restructured several paragraphs and removed those on autophagy and apoptosis which did not receive equal weight and detracted from our main message. We believe this has clarified the most recent developments (summarized in Fig. 1 and Fig. 2) in the field of immunometabolism during staphylococcal infection.
Abbreviation should be explained when first appeared in the text.way to S. aureus infection, focusing on the immunometabolic responses. We have ensured that all abbreviations have been explained when they first appeared in the text (lines 37 and 40).
Round 2
Reviewer 2 Report
This referee is very satisfied by the changes made by the author(s).
The manuscript reads now extremely well, with a logical flow and excellent insights in the field.
I strongly recommend the publication of this review, which will attract a broad readership and will certainly be very well cited.
Congratulations to the author(s) for doing such a stellar work in improving this manuscript to such a degree of quality. It was a pleasure to review it.